# Autonomous Vision-Based Primary Distribution Systems Porcelain Insulators Inspection Using UAVs

**DOI:** 10.3390/s21030974

**Published:** 2021-02-02

**Authors:** Ehab Ur Rahman, Yihong Zhang, Sohail Ahmad, Hafiz Ishfaq Ahmad, Sayed Jobaer

**Affiliations:** 1College of Information Science and Technology, Engineering Research Center of Digitized Textile & Fashion Technology, Ministry of Education, Donghua University, Shanghai 201620, China; 318011@mail.dhu.edu.cn (E.U.R.); 318098@mail.dhu.edu.cn (S.A.); 318058@mail.dhu.edu.cn (S.J.); 2School of Computing, Faculty of Engineering, Universiti Teknologi Malaysia, Johor 79100, Malaysia; hiahmad2@graduate.utm.my

**Keywords:** primary distribution systems, transfer learning, YoloV4, porcelain insulator detection, UAVs, BRISQUE, LIME, LapSRN, YoloV5

## Abstract

The early detection of damaged (partially broken) outdoor insulators in primary distribution systems is of paramount importance for continuous electricity supply and public safety. Unmanned aerial vehicles (UAVs) present a safer, autonomous, and efficient way to examine the power system components without closing the power distribution system. In this work, a novel dataset is designed by capturing real images using UAVs and manually generated images collected to overcome the data insufficiency problem. A deep Laplacian pyramid-based super-resolution network is implemented to reconstruct high-resolution training images. To improve the visibility of low-light images, a low-light image enhancement technique is used for the robust exposure correction of the training images. A different fine-tuning strategy is implemented for fine-tuning the object detection model to increase detection accuracy for the specific faulty insulators. Several flight path strategies are proposed to overcome the shuttering effect of insulators, along with providing a less complex and time- and energy-efficient approach for capturing a video stream of the power system components. The performance of different object detection models is presented for selecting the most suitable one for fine-tuning on the specific faulty insulator dataset. For the detection of damaged insulators, our proposed method achieved an F1-score of 0.81 and 0.77 on two different datasets and presents a simple and more efficient flight strategy. Our approach is based on real aerial inspection of in-service porcelain insulators by extensive evaluation of several video sequences showing robust fault recognition and diagnostic capabilities. Our approach is demonstrated on data acquired by a drone in Swat, Pakistan.

## 1. Introduction

Low-voltage power distribution lines are the means of electricity distribution from the distribution grid to the end users. An important aspect of a primary distribution system is a continuous supply of electricity and the efficient performance of its equipment. Insulator strings are essential equipment in primary overhead power distribution lines because of their role in insulation and providing mechanical strength. An insulator’s efficiency is affected when exposed to pollution, environmental conditions such as dust, rain, wind, or snowfall, and wildlife. Components of such importance cause serious problems when damaged, both to the power supply and public safety. Every year, a number of human lives are taken due to electric shocks during the rainy season due to exposure to electric poles. In 2019, in the rainy season in Karachi, Pakistan, six people died due to electric shocks due to poor insulation of the distribution lines from the electric poles [1]. In order to prevent such severe and costly damage, periodic maintenance and detection of defects in the early stages are of great importance. Some of the defects present in primary distribution systems are broken insulators, broken cross arms, conductor corrosion, and vibration damage. Some normal and defective insulators from different view angles are shown in Figure 1. An illustration of the overhead primary power distribution lines is depicted in Figure 2.

Research on porcelain insulators present on electric poles for supporting and insulating the overhead lines is very scarce. This might be due to the non-availability of such systems in developed countries or limited resources in developing countries for researching these systems. One of the objectives of this study is to shed some light on these systems. In Pakistan, overhead power distribution lines are commonly present both in urban areas and the countryside. In these systems, porcelain-type insulators are commonly used due to their robustness and low cost. The pin-type and suspension disc-type porcelain insulators are used for a voltage range of 11 kV to 33 kV. Such a pin insulator sits on the cross arm of the electric pole, which has grooves on the upper end to hold the conductor and insulate the electric pole from the conductor. The suspension disc insulator provides mechanical strength and insulation to power lines. The rain shed or petticoats made of porcelain, a non-porous and waterproof material present on both pin and disc insulators, provide a long leakage path to avoid flashovers and puncture. A broken rain shed poses a risk of flashovers and outages of power, which needs to be detected in advance to prevent future anomalies.

Several researchers employed methods to inspect, detect, and analyze such defects using computer vision techniques. Van et al. presented an extensive review on some of the current methods and techniques to inspect, identify, and classify such defects in power equipment mounted on electric poles, along with different weather conditions, using computer vision-based techniques [2]. Adrian et al. discussed different applications of deep learning in unmanned aerial vehicles (UAVs) along with performances and limitations [3]. Thus, to provide safety to the public and maintenance crews and continuous delivery of power to end users, the prevention of such defects is a top priority of electrical companies. To prevent the aforementioned defects, a periodic inspection is carried out by electrical companies.

Two common methods of power system inspection include maintenance crew personnel patrolling by foot, examining each component. Such patrolling is costly, risky, and time-consuming. Another patrol method is by manned helicopters that fly a safe distance from the power lines and equipment and a cameraman records the video of these systems for later investigation. Such patrolling is fast, expensive, risky, and less accurate. In recent research, Xie et al. used a large unmanned helicopter along with multiple sensor data for power line inspection [4]. This system is fast and inexpensive but further optimization needs to improve the accuracy and success rate. To perform fast and quality inspections, mobile robots such as climbing robots have been used. Such climbing robots move along the conductor for inspection. Jaka et al. surveyed such robots and presented their main characteristics [5]. To reduce maintenance costs and mitigate downtime and emergency repairs, Rebecca et al. designed and tested a power line robotic device as a tool for maintenance crews for preventive inspection [6]. Certain sensors, like a camera LiDAR and GPS chip, can be embedded in the climbing robot to analyze the power lines to improve the accuracy of the inspection results. Xinyan et al. proposed a cable inspection robot for the automatic inspection of transmission lines to reduce manpower and improvement of inspection accuracy [7]. Recently, unmanned aerial vehicles have been used for different applications in many areas, such as surveillance, security, and inspection. High operational costs and security concerns urged electric utility companies to utilize UAVs for the inspection and maintenance of power equipment. The regular advancement of automatic flight controls and more efficient computer vision-based detection, classification, and tracking techniques allow UAVs to perform low-cost and efficient inspections of power equipment from a safe distance. Based on different data types, such as GPS data, visible and infrared images, and LiDAR data, UAVs provide high performance in inspection tasks [8,9,10,11,12,13].

Obstacle avoidance and path planning is another major point of research on UAVs reduce human interference during its operation and provide fully autonomous navigation. Classically simultaneous localization and mapping (SLAM) systems use data taken from LiDAR and RGB-D cameras to infer the visual geometry of obstacles and spaces, which leads to achieving obstacle and collision avoidance [14,15]. Such systems are highly expensive and an alternative for such a system is a computer vision algorithm used for depth estimation or optical flow from the data taken from a stereo camera [16,17]. Such a system is more cost effective than the latter sensors but needs more computational power. Further research is needed in such regard to make such a system less complex and to reduce the overall consumption of computational resources.

Path planning is another major concern in operating a drone as it ensures autonomous flight and the optimal path of flight for the UAV to reach its goal. The latest evolving fuzzy algorithms and controllers for the path optimization and control of UAVs and multi-copters have been reviewed [18]. Yang et al. developed a famous metaheuristic firefly algorithm [19] and the work was inspired by natural firefly flashing behavior. Another study addressed a continuous optimization problem by proposing a hybrid particle swarm optimization along with a firefly algorithm [20]. Thus, for a better solution, hybrid fuzzy and firefly algorithms acquire the features of both controllers to achieve an optimal solution for time and path planning [21].

Super-resolution reconstruction is a technique developed to improve the quality of low-resolution (LR) images and reconstruct high-resolution (HR) images. Traditionally, the upscaling of the image was based on the interpolation of nearby pixels and taking a mean average of pixel values by adding another pixel. Such upscaling methods reduced the overall quality of the image during the reconstruction process. Deep learning-based super-resolution techniques addressed this problem by different techniques to achieve high peak signal to noise ratios. In [22], to improve the quality of images in the training dataset, a super-resolution convolutional neural network is implemented on blurry images taken by a drone. Such methods are continuously evolving and new algorithms with high efficiency are being developed to improve the overall output of such networks. Deploying such advanced deep learning-based super-resolution neural networks will improve the overall performance.

Critical to many object detection and object tracking applications, the high visibility of image features is of paramount importance. Images taken in low-light conditions shutters some of the details, reducing the quality and visibility. Low-light enhancement algorithms proved successful on such images, improving the visibility and quality of the image [23,24]. Aimed at the quantitative representation of human perception of image quality, quality assessment algorithms have been developed. These algorithms are divided into two subclasses, including a high-quality reference image used to evaluate two images [25]. An image quality assessment (BRISQUE) describes the image structure by calculating features and a human opinion of image quality based on those features [26]. Such methods are utilized to improve the training and validation datasets and the overall object detection and training accuracy.

Compared to conventional methods, UAV-based inspections, along with computer vision techniques, provide safer, less expensive, and more robust inspections of power system equipment. However, still, a lot of discrepancies are found in these systems, such as low endurance. Robust algorithms for object avoidance and more efficient and accurate computer vision-based object detection techniques, etc., are still needed. In particular, a fully autonomous system of UAVs to automate the whole inspection process is still far from reality. Such a system needs a highly trained human pilot to fly the drone at a safe distance from the electric lines to protect the UAV from any hazards. A video from a drone camera is analyzed frame by frame by an object detection model to assess the maintenance status of porcelain insulators. Additionally, autonomous UAV-based inspection provides a safer inspection from a distance above the overhead power lines, capturing each component of the power system using different kinds of sensor for further investigation.

In the present work, we aim to achieve the aforementioned objectives and focus our work on faulty porcelain insulator detection in a low-voltage power distribution system in which insulators provide mechanical support to overhead power distribution lines and insulate them from electric poles. We present our work as follows:We demonstrated and developed an algorithm for improved UAV-based low-voltage porcelain insulator inspection which is commonly used in Pakistan in the power distribution system. Among the several components present in the power system infrastructure, we devoted our study to examining the structure anomalies and faults present in pin and suspension disc insulators insulating and supporting the power cables from the electric towers.Regarding the structure anomalies, we developed a novel dataset of visible insulator images by preprocessing the training and validation datasets (increasing dataset) along with a Laplacian pyramid-based super-resolution network (LapSRN) to acquire high-resolution images and improve the image quality.Low-light enhancement and exposure correction for less visible images. These different low-light images with varying lambda and gamma values are used as an augmentation technique for the training dataset. The images after processing through this pipeline were evaluated with the blind no-reference image quality assessment (BRISQUE) to prove their improved quality and visibility.A different fine-tuning strategy was implemented in state-of-the-art object detector YoloV4 for improved detection accuracy for fault detection and classification.We automated the inspection and for improved inspection accuracy requiring less flight time, we developed a path planning strategy for the UAV flight.The proposed method was evaluated with a test dataset comprising different backgrounds, light conditions, and complex scenarios.

Other sections of the paper are presented in the following manner. Section 2 describes the background knowledge of all the methods and techniques used to improve the efficiency of detecting and classifying insulators and our proposed approach. Section 3 presents the experimentation, comparison of different object detection models, flight path strategies, and the results achieved. Finally, Section 4 concludes the achieved results and details some future work directions.

## 2. Materials and Methods

The surge in the usage of UAV technology has recently been boosted in numerous fields, such as agriculture, wildlife surveillance, search and rescue operations, and also the power sector. A need for more advanced and robust object tracking and detection algorithms has emerged. Additionally, with UAVs that have a long endurance time and are robust to environmental changes, fast, and automatic, less human support is needed. In the power sector, high-voltage power lines need automatic inspection using UAV intelligence and support to reduce the risk to the maintenance crew. A fully automatic scheme where a UAV does not need any human pilot for its operation would provide safe and fast operative maintenance. Such algorithms need to be developed. In our research, we developed a prototype to accomplish the abovementioned objectives, which include: (i) A fine-tuned efficient object detector and tracker for insulator fault detection and classification and (ii) a flight path planning strategy for the drone to fly and inspect insulators from one electric pole to another with minimal human support.

In this regard, we devoted our work to the detection of broken insulators of both pin and suspension disc types from image sequences acquired by a drone camera carrying out inspection from one electric pole to another. To detect faults in the insulators, the first step is to preprocess the data for the efficient training of the object detector and classifier. The dataset of the faulty insulators is enlarged by adding different types of faulty insulators from different angles of view and different background images taken by a drone camera. In the second step, different image-processing algorithms are used to improve the quality of the dataset to differentiate between insulator types. We then utilize visible images to extract normal and faulty insulators. In the third step, we define different flight path patterns to efficiently investigate the porcelain insulators mounted on an electric pole. The framework of the proposed methodology is presented in Figure 3. In the following sections, we present a brief discussion of the techniques used to inspect a video stream coming from a UAV camera for the detection and classification of faults in low-voltage porcelain insulators from visible RGB images, along with flight path patterns for the drone.

### 2.1. Framework of Insulator Detection and Classification

The proposed deep learning-based low-voltage porcelain detector consists of three stages. Data augmentation and image preprocessing, training and validation of both real and custom YoloV4 architecture, UAV path planning strategy, and real-time insulator detection. The framework for the low-voltage insulator detection is depicted in Figure 3**.** First, images of insulators, both normal and faulty pin and suspension disc types, are collected using a UAV, a digital camera, and a mobile phone for distribution line inspection. Different datasets are used in the fine-tuning and validation of the object detection model. The preprocessing of these datasets includes dataset generation, data augmentation, image reconstruction using LapSRN, low-light enhancement, image quality assessment, and dataset labeling. For the training stage, we utilized YoloV4 architecture by feeding in the training images. A normal insulator dataset is fed into the YoloV4 architecture in stage A and transfer learned with normal insulator features following stage B in which some of the previous layers are frozen and the detection layers are fine-tuned with the faulty insulators dataset. Such a strategy proved efficient in the detection of the faulty insulators in the aerial images and frames taken by the UAV for the distribution line inspection system.

### 2.2. Electric Poles and their Components

In low-voltage power distribution lines, electric poles consist of different components for specific tasks. The inspection and maintenance of such components are of paramount importance for the continued supply of power to the end users. Due to the presence of these systems in urban and congested areas, public safety is a major concern. Most electric poles serve as a mechanical support for different components present in these distribution systems. Such components comprise insulators, transformers, three-phase ACSR wires, cross arms, etc., shown in Figure 4.

Such systems support 11kV three-phase wires distributed from pole to pole. At each utility feeder location, there are step-down transformers that step down the voltage from 11 kV to 220–240 V for consumers. Pin-type insulators are used to provide insulation to the three-phase wires from the electric pole and provide support for the distribution wires. Disc insulators are used for mechanical strength and supporting endpoints of the conducting wires. Different kinds of fault are present in such insulators, which mostly interrupt the power supply and cause a risk to the public. Such faults are shown in Figure 5.

The internal structure of pin and disc insulators is shown in Figure 6. The rain shed is a source of insulation between the pole and the conducting wire. The rain shed increases the insulation distance between the pole and the wire. When the rain shed is broken, the insulation distance decreases and, due to the arcing effect or air ionization effect, a connection between the pole and wire is established, leading to high current flow and puncturing of the insulator and shutdown of the power supply. Such a scenario is known as flashover. Power supply companies need to identify this fault in advance and replace the respective component in time, and strategies need to be developed for the power supply company to overcome such problems beforehand. Reducing the manpower used for inspections is a time-consuming, risky, and costly procedure. Using UAVs and the proposed object detection techniques can play a major role in reducing such problems, increasing inspection and maintenance quality with fewer risks.

### 2.3. Image Preprocessing

Images of porcelain insulators mounted on the 11 kV electric poles in the visible spectrum were acquired from video frames taken from a drone, a Canon DSLR camera, and a mobile phone in Mingora City, Swat (Pakistan) in June 2020. To ensure the safe acquisition of images of the electric poles, the safety procedures for power-operated tools with 11 kV ACSR lines were followed. The safe distance from power lines is 3 m to avoid electromagnetic interference from the power-operated tools [27]. Most of the images were taken of the overhead power lines present in commercial as well as residential areas. Based on different angles of views and light exposure, the training dataset, as well as a testing dataset, were acquired. Figure 7 presents some of the images taken from different view angles.

The dataset is divided into four classes, including pin insulator, disc insulator, faulty pin, and faulty disc insulator. To differentiate between the faulty (broken) insulators and normal (intact) insulators, insulators with a partly broken rain shed or a small piece broken off so that the white porcelain material is visible and insulators that are deformed are compared to a corresponding normal intact insulator. These faults could easily create a risk for pollution deposition and surface corrosion. Such different faulty (broken) insulators are shown in Figure 7b. The proportion of faulty insulators in the dataset is very small due to the unavailability of such insulators in the system. Such insulators are discarded by throwing them in the garbage as no proper method is available for their disposal. To overcome this problem and to increase such images in the training dataset, we followed the study in [28] to improve our dataset and remove the class imbalance. Frames are extracted from drone videos comprising different backgrounds and different light conditions. Faulty insulators from a different angles of view are then added to these backgrounds extracted from real images. Figure 8 shows the scenario for such dataset formation step by step.

Another set of images and videos was made for evaluating the detection, classification, and fault diagnosis of insulators. This dataset was mainly acquired using a UAV with different flight path patterns with a prototype of an electric pole on which a faulty disc and pin insulator were mounted. These experiments were performed in a controlled environment. Figure 9 shows the UAV, its technical specification, and the prototype of the electric pole used for testing our proposed method for insulator fault diagnosis.

### 2.4. Image Reconstruction Using LapSRN

To improve the insulator fault detection performance the data acquired from UAVs and other sources, images should be of high quality and have rich features. During the acquisition of the training dataset, some images were blurred. Another problem was cropping the faulty insulator patch from images with multiple insulators with both normal and faulty pins and suspension discs, as shown in Figure 10. Cropping the image degrades the resolution of the images. Another problem was distortion in images due to UAV vibration and fuselage. To overcome these problems, we used LapSRN. Super-resolution techniques with deep Laplacian pyramids provide a fast and accurate super-resolution solution for high-resolution image reconstruction. Such a technique allows for reconstructing the low-resolution patches of faulty insulators taken from images consisting of both normal and faulty insulators. LapSRN is comprises two stages. A bundle of convolutional layers learns a non-linear feature map from the low-resolution input image in the feature extraction branch and upsamples the low-resolution image to a finer level in the image reconstruction branch, then with the help of convolutional layers, the residuals are predicted [29]. The overall workflow of the LapSRN algorithm on low-resolution images is shown in Figure 10.In the first step, there is a transformation of high-dimensional non-linear feature maps by feature embedding networks.In the second step, there is an upsampling of the extracted features by transposed convolutional layers by a scale of 2.Lastly, a sub-band residual image is created by convolutional layers ( ${\mathrm{Conv}}_{\mathrm{res}}$).

In the training of LapSRN in all convolutional layers except the first layer, 64 filters are used on the input LR image, image upsampling layer, and layers for predicting residuals. A filter for the convolutional layer and transposed convolutional layers of size 3 × 3 and 4 × 4 are used, respectively [29]. To keep the input of each level the same size, a zero-padding step around the boundaries is used before convolution. To generate an HR image $\hat{y}=f\left(x;\theta \right)$ a mapping function similar to the ground truth y is learned. Instead of minimizing the mean square errors (MSEs) between ${\hat{y}}_{l}^{\left(i\right)}\;and\;y_{l}^{\left(i\right)}$, an efficient loss function to handle outliers is used. The loss function is calculated as
(1)$$L_{s}\left(y,\hat{y};\theta \right)=\;\frac{1}{N}\sum_{i=1}^{N}\sum_{l=1}^{L}\rho ((y_{l}^{\left(i\right)}-\;x_{l}^{\left(i\right)})\;-\;{\hat{r}}_{l}{}^{\left(i\right)})\;$$

From Equation (1), θ is the set of parameters in the network to be optimized, at each level ι the residual image is represented as ${\hat{r}}_{l}$, as shown in Figure 10a, and the HR image is represented by ${\hat{y}}_{\iota }$ and the corresponding LR image by $x_{\iota }$. Additionally, the L1 norm differential variant known as the Charbonnier penalty function is represented by $\rho \left(x\right)=\;\left(\sqrt{x^{2}+\in^{2}}\right)$, S presents the scaling factor of upsampling, and the number of training samples is presented by N. The pyramid levels in the model are represented by $L=\log_{2}S$.

### 2.5. Low-Light Image Enhancement

During the acquisition of the training dataset, there were some images with a low-light problem due to the shadow of a nearby building or due to cloudy weather. Such conditions can hinder a lot of features of the insulator, especially the color, making it difficult to differentiate between a black and a dark brown insulator. Such a problem in the training dataset may lead to low visibility of the insulators and their structure. Such artifacts present in the images degrade the overall performance of computer vision and image-processing techniques. To overcome these problems, we used the algorithm of [23] to enhance the low-light images for exposure correction. LIME is an algorithm that enhances the exposure of an image by illumination map estimation.

LIME is built on the retinex model, showing the development of a low-light image. It is
(2)L = Τ o R
where *L* is the captured image, *T* represents the illumination map, and *R* is the recovery desired for *L*. Element-wise multiplication is represent by the symbol “*o*”.

The LIME algorithms takes low-light input image parameters with a positive coefficient and gamma corrections are designated. A weight matrix is key to designing the initial illumination map for the structure-aware refinement. Such a weight matrix is calculated by Equation (3).
(3)$$W\upsilon \left(x\right)\leftarrow \sum_{y\in \Omega \left(x\right)}\frac{G\sigma \left(x,y\right)}{\sum_{y\in \Omega \left(x\right)}\sigma_{v}\left(x,y\right)\nabla \nu \hat{\hat{T}\left(y\right)+\epsilon }}$$
where Gσ(x,y) is formed by the Gaussian kernel by using the standard deviation σ and $\nabla v\hat{T}$ is the first-order derivative filter. Additionally, ϵ is a small constant value to prevent zero denominators. Gσ(x,y) is calculated as
(4)$$G\sigma \left(x,y\right)\;\propto exp\left(-\frac{dist\left(x,y\right)}{2\sigma^{2}}\;\right)$$
where the spatial Euclidean distance between location x and y is represented by the function dist(x,y). In Equation (4), the initial illumination map is estimated for each low-light input image pixel by Equation (5):(5)$$\hat{T}\left(x\right)\;\leftarrow max_{c\in \left\{R,G,B\right\}\;}L^{c}\left(x\right),$$

In Equation (5) x represents each individual pixel of each channel and L represents the low-light image. The initial illumination map is then refined from $\hat{T}$ to **T** by sped-up solver Equation (6):(6)$$\left(I+\;\sum_{d\in \left(u,v\right)}D_{d}^{T}\;Diag\left({\tilde{w}}_{d}\right)D_{d}\right)t=\;\hat{t},$$
where I is the identity matrix and Equation (6) represents a symmetric positive definite Laplacian matrix with Diag(x) to compute the diagonal matrix by using vector x. Gamma correction is applied to the refined illumination map **T** and the low-light input image **L** is enhanced by the application of Equation (1). Furthermore, if recomposing and denoising is needed, then the denoised and recomposed output image is acquired by Equation (7):(7)$$R_{f}\;\leftarrow R\;o\;T+\;R_{d}\;o\;\left(1-T\right),$$
where $R_{d}$ and $R_{f}$ are the output results after recomposing and denoising. A final enhanced image is acquired by the use of the above process. Some of the results obtained on different training images are shown in Figure 11. Such variations of different gamma and lambda values are used as an augmentation technique to increase the detection and generalization of the insulator detection model in different light conditions.

### 2.6. Image Quality Assessment

Image quality is one of the center points for the high performance of object detection models, as a highly visible image provides rich spatial features of the objects present in the image. In view of such requirements, we used in this work a quality assessment on part of the training images, including different image processing techniques, such as Laplacian-based super-resolution on low-resolution images and low-light enhancement on low visibility in certain images. To evaluate the quality of such images to be suitable for training the insulator detection model, we used image quality assessment algorithms.

Image quality assessment algorithms are quantifying metrics used to observe and evaluate the performance of different computer vision and pattern recognition tasks, such as image processing, image compression, and image transmission [30]. These algorithms comprise two groups, i.e., a reference-based quality assessment in which a high-quality image is taken as a reference and a distorted or low-quality image is compared to it. Another type that is utilized in our work is BRISQUE [26], a blind reference image spatial quality evaluation. This algorithm is highly efficient compared to the reference-based evaluation as it does not require any transformation for calculating features from image pixels.

To calculate the BRISQUE value for a given image, first, mean subtracted contrast normalized (MSCN) coefficients also known as locally normalized luminance, are calculated by Equation (8) [31]:(8)$$\hat{I}\left(i,j\right)=\;\frac{I\left(i,j\right)-\;\mu \left(i,j\right)}{\sigma \left(i,j\right)+C}$$

In Equation (8), $\hat{I}\left(i,j\right)$ computes locally normalized luminance for a given image I(i,j) by using local mean subtraction μ(i,j) divided by local deviation σ(i,j), and C is a constant to avoid zero division. A better fit to the empirical histogram for the coefficient products is represented by an asymmetric generalized Gaussian distribution [31]. The fitting of MSCN coefficients to generalized Gaussian distributions and the pair-wise products of the asymmetric generalized Gaussian distribution produces the resultant features needed to calculate the image quality. The image quality after implementing BRISQUE is calculated on a scale of 1–100. The lower the output scores, the better the quality of the image.

### 2.7. YoloV4

Yolo (you only look once) is a single-stage detector designed for real-time object detection which performs object classification and localization at the same time. In object detection, high real-time processing frame rates and detection accuracy are the primary objective. The YoloV4 object detection model is benchmarked on the MS COCO dataset [32], achieving 65 fps inference speed with an accuracy of 43.5% AP (65.7% AP_50_) on Tesla V100 [33]. Object detectors compress features of an input image down through a convolutional neural network backbone. The mixing and holding up of the feature layers from the convolutional backbone happens in the neck part of the object detector. The detection of a specific object in the image happens in the head part of the detector. As YoloV4 is a single-stage object detector, the classification and prediction of object localization are done at the same time.

#### 2.7.1. Backbone

The backbone of YoloV4 is based on CSPDarknet53. The convolutional architecture is based on a modified DenseNet [34]. The edited DenseNet uses cross-stage partial connections that send one copy of the feature map separated from the base layer through the dense block and another to the next stage. The major advantages of choosing DenseNet architecture are alleviating the gradient vanishing problem, bolstering backpropagation, and fewer network parameters, while, when using the cross-stage partial connections, the computational bottleneck of DenseNet is removed, with improved learning.

#### 2.7.2. Neck

Feature aggregation occurs in the neck part of the YoloV4 object detector. Path aggregation networks (PANets) are used by the YoloV4 detector for feature aggregation along with a spatial pyramid pooling block after CSPDarknet53 to increase and improve the receptive field and sort out the most important features from the backbone.

#### 2.7.3. Head

Anchor-based detection steps are deployed with three levels of detection granularity in the head region of the YoloV4 detector, the same as those implemented in YoloV3 [35], a previous version of Yolo. Certain novel features have been added to YoloV4, such as bag of freebies, which includes different augmentation techniques, drop block regularization, complete IoU loss (CIoU), etc. Additionally, some bags of specials are also included that consist of mish activation, DioU-NMS, modified path aggregation networks, etc. [33]. The complete structural diagram, with different blocks of YoloV4, is presented in Appendix A
Figure A1.

### 2.8. Proposed Fine-Tuning Strategy for YoloV4

To overcome the class imbalance problem and scarce dataset of faulty insulator images, we propose a different fine-tuning strategy of insulator defect detectors for efficient training. To achieve the best results on object detection tasks, deep learning models need enormous amounts of training data. Such enormous amounts of data are needed for the object detectors to learn features of the specific object of interest so that the model generalizes well when exposed to unseen test data. Recently, different techniques have been developed to overcome data insufficiency domain gap problems. These techniques include data augmentation in which the labeled training samples go through different procedures, such as geometric juxtaposition, cropping, scaling, mosaic, flipping and zoom augmentations, etc. Such techniques increase the richness of features learned by the deep learning architectures. The insulators that are the focus of this work have no available open-source dataset, and with small datasets, the available data augmentation techniques do not overcome the problem of the overfitting of deep convolutional architectures which decreases the detection accuracy of the object detection models.

To overcome such limitations, we fine-tuned the insulator detection model in two steps. In step one, a state-of-the-art model YoloV4 [33] pre-trained on the MS COCO dataset is considered for fine-tuning insulator detection. The COCO dataset consists of 2 million images and 80 object classes. These classes do not contain the insulators that are the focus of this work. However, such learned features from those 2 million images can be used for better detection accuracy of the insulator detection model. For that reason, in step two, we transfer learned normal insulator images on top of the previously learned features from the COCO dataset so that the model learns different insulator features present in the training samples of the normal insulator dataset. The step-wise procedure of the two-step fine-tuning strategy is depicted in Figure 12.

The main focus of electric supply companies is to inspect defective and broken insulators to ensure safe and reliable power delivery to consumers. Low-power distribution systems are mostly present in densely populated areas, and the insulation of such power supply apparatus is of paramount importance for public safety. That is why faulty and defective insulators are the major concern for electric supply companies. In light of this, the insulator detection model should be more robust to find these faulty and broken insulators in advance. For that reason, a second fine-tuning strategy is performed. In the second step, the previous single-step fine-tuned model parameters are again fine-tuned with the training samples containing only defective insulators labeled as a faulty pin and faulty disc insulators. Upon fine-tuning, the features from the defective insulator dataset which are more meaningful in terms of defective insulators are learned by the detection model. Using a two-step strategy increases the accuracy of the detection model in detecting specific faulty insulators, and also resolving overfitting and data insufficiency problems.

### 2.9. Drone and Flight Path Trajectory

To improve the maintenance crew’s safety, power system components mounted on electric poles can be inspected with the use of drones. In this study, to perform our experiments, we developed a prototype of an electric pole, as shown in Figure 9, on which the faulty pin and disc porcelain insulators are mounted and examined using a DJI Tello drone [36], which is a programmable drone. The technical specification of the drone is presented in Table 1. The drone camera captures 5 megapixel stills with a field of view of 82.6° and records 720 p HD video footage at 30 fps in MP4 format. Frames of the video have been examined by the insulator detection model and the results are presented in the results section.

Most of the previous works on insulator detection lack a complete explanation of the process of taking the video feed with the drone’s camera. Research on low-voltage power distribution systems is scarce but is highly important, as these systems are present in densely populated areas. To automate the process of inspection, in this work, we defined different flight path strategies and captured the frames of pin and disc insulators from different orientations. These different flight path strategies are important to overcome the shuttering of the insulators due to the presence of conductors and the top tie clipping wires. The clipping wire is used to tie the conductor wire on the head of the pin insulators and to the central pin of the disc insulator. Such a shuttering phenomenon is shown in Appendix A
Figure A2a. Focusing on this problem of how to avoid these shuttering phenomena, different flight path strategies are defined and the effects of such strategies are evaluated. These strategies are divided into two types, as shown in Figure 13.

In Figure 13a, a 270° rotational path is implemented in the drone around the electric pole and a video is recorded consisting of frames including faulty pin and disc insulators. The drone is again flown in the straight direction, as depicted in Figure 13b, hovering above the power lines at a safe distance and upon crossing the cross arm of the electric pole from above, the drone itself is rotated by 180° to take a view of the back side of the cross arm and components mounted on the electric pole. The frames have a complete view of the top, front, and back sides of the insulators. The major advantage of this strategy is that it is more energy- and time-efficient for the drone. This strategy is a lot safer due to the safe flight path provided by the width of power lines, as shown in Figure 14. The safety limits of the powered device with electric poles were considered while performing the above experiments and such safety guidelines were described earlier.

Due to congestion and close proximities of buildings and trees near the electric poles, applying circular flight trajectory will be difficult and needs more complex sensors and control theory. The straight path for the drone as devised in this study is helpful in a less complex flight path. The distance between the three-phase conductors is enough for the drone to have a smooth flight path and avoid the buildings and surrounding objects. The field of view of the camera can capture the front, back, and top sides of the insulators and they can easily be inspected.

## 3. Experimentation and Results

### 3.1. Data Preparation

For our model, we gathered aerial images taken by the UAVs from PESCO Pakistan low-voltage distribution lines and power systems [37]. For training, we gathered two different novel datasets, as shown in Table 2, one for training and one for validation of a single-stage YoloV4 object detection model. This dataset consists of 5939 images containing both faulty and normal pin and disc insulator images. For training the custom YoloV4 architecture, in the first step, the first 135 layers were frozen and the rest were fine-tuned on 4827 images containing only normal pin and disc insulator images. In the second step, the detection layers of the YoloV4 architecture were fine-tuned with 4099 images containing faulty pin and disc insulators only. For evaluating the single-stage YoloV4 detection model, three different test datasets were used. Furthermore, to evaluate our two-step fine-tuned insulator detection model, two different test datasets were used. The details of the training and testing datasets are presented in Table 2.

R-1 consists of images taken from a UAV of normal insulators mounted on an electric pole. These images were taken in a low-light condition, making it harder for the detection model to generalize. R-2 consists of images also taken from a UAV of normal insulators mounted on an electric pole but with better light conditions and orientations. The complex (C) dataset consists of images taken from a still camera and consists of images of the electric pole with a larger number of insulators of both types with shuttering effects due to conductors and tying wires. Such a dataset is used as a stress test on the insulator detection model to prove its generalization ability in complex scenarios. Prototype-C contains video frames that are taken from the prototype electric pole on which faulty suspension disc and pin insulators were mounted. This dataset was recorded during the circular flight of the UAV. Prototype-P consists of the frames taken by the UAV with the proposed flight path for insulator inspection. Most of these datasets consist of multiple objects per image. Additionally, different images have different backgrounds, such as blue sky, cloudy sky, vegetative background, scenes containing buildings, and also complex backgrounds. Example images of different training datasets are shown in Appendix A
Figure A2 and samples from testing datasets are shown in Appendix A
Figure A3. All the datasets are manually labeled with ground truth boxes. Among these datasets, each contains a random number of both pin and disc insulators of both normal and faulty types.

### 3.2. Implementation

The proposed insulator detection model is implemented in the Darknet framework with object detection API. The training of the insulator detection model is powered by the Google Colab cloud platform. The hardware is equipped with 12.6 GB RAM and a 12 GB Tesla K80 GPU. We used the Windows 10 platform. For evaluation, we also used a computer equipped with NVIDIA GeForce GTX 1060 GPU with 16 GB of RAM, Intel Core i5-8400 CPU 2.80 GHz (6 CPUs), on the Windows 10 platform. The trained weights were converted to the Tensorflow [38] object detection API model to check the test videos. To acquire the test dataset, we semi-autonomously controlled the UAV by writing Python libraries, such as pygame and Tello drone blocks. We also used the DJI Tello app for recording the videos and performing different flight patterns. The video frames taken were then tested with the saved Tensorflow model weights. For the deep Laplacian pyramid, super-resolution networks and low-light estimation techniques for image enhancement were also implemented in the Tensorflow framework with the same hardware setup as stated earlier.

The parameters for fine-tuning the insulator detection model were set as follows: the initial learning rate is 0.0013 with decay 0.0005 and momentum is set to 0.949. A 64 batch size with 32 subdivisions. Cross iteration batch normalization (CmBN) is applied to prevent overfitting and noisy estimations. The IoU threshold is 0.213 and complete IoU loss (CIoU) is selected as the IoU loss function to minimize the central point distance of the predicted box and the ground truth box, to maintain consistency of the box aspect ratio, and to increase the overlapping area [39]. Precision, recall, precision–recall curve (PRC), F1-score, and average precision (AP) are metrics selected to evaluate the detection performance of the model.

### 3.3. Insulator Detection Results

#### 3.3.1. Image Quality Assessment for Specific Insulator Dataset

Faulty insulator inspection and maintenance are the top priority for electricity supply companies to maintain uninterrupted power supply to consumers. The need for efficient fault detection models is immense. For such requirements, we need to improve the quality of the training dataset and provide a highly efficient faulty insulator detection model which is accurate, fast, and provides reliable results. For this reason, as shown in Figure 15, a pipeline has been proposed in which a low-resolution image of 320 × 180 is fed into the super-resolution network based on deep Laplacian pyramids. The output of LapSRN is upsampled by a scale factor of 4. Such super-resolution networks preserve the quality better as compared to the traditional upscaling techniques. The image quality score is assessed by using blind no-reference image spatial quality estimation to prove the improvement in the quality of the image. For some of the training images, such as those with low visibility, the LIME technique is applied. Such an algorithm is efficient in making the object’s features in the image clearer and helps in learning better features in the detection model. After the implementation of LIME, again, the image quality score is calculated to prove the feature improvement of low-light images. As shown in Figure 15, an improvement of 12.1 points is achieved in the quality of the image. Different variants of the single low-light image can be obtained by optimizing various gamma and lambda values of the LIME algorithm. The obtained images can possibly be used as a novel augmentation technique. The scenario is depicted in Appendix A
Figure A3f. Such quality improvements are varied based on image distortion correction and low-light enhancement. The respective plots in Figure 15 present the distribution of the mean subtracted contrast normalized (MSCN) coefficients in different directions, such as vertical, horizontal, main diagonal, and secondary diagonal. Such distributions with a regular structure present a better quality and, as shown in Figure 15, the distribution is more regular after implementing the LIME algorithm.

#### 3.3.2. Normal Insulator Detection in Primary Distribution Power Systems

To check the viability of the YoloV4 architecture and the tuning parameters, along with the training dataset, we first trained YoloV4 and its variant YoloV4 Tiny for the insulator detection model. The testing results for different datasets of both architectures are presented in Table 3. All the testing datasets comprise only normal pin and suspension disc insulators. We took into account a different conditions of these test datasets to check the generalization ability of the normal insulator detection model. The test set R-1 had frames taken in a low-light condition by the UAV of a current in-service electric pole.

YoloV4 presents an average precision (AP) of 82.9% in both classes, including an AP of 78.2% on pin insulators and 87.7% on suspension disc insulators. As YoloV4 Tiny is a light model with high inference speed, the AP is lower as compared to the full YoloV4 model. Another test set, R-2, is composed of video frames taken by the UAV in better light conditions where all the components are highly visible. An AP of 99.5% on pin insulators and 95.2% on disc insulators is achieved. These results are better than the R-1 dataset due to the high visibility of insulators. The complex dataset is composed of images taken from a mobile phone camera of different complex electric poles present outside a 132 kV grid substation. Such electric poles have a large number of components per electric pole, as shown in Appendix A
Figure A2. An AP of 81.5% is achieved on pin insulators and 83.8% on suspension disc insulators. The results with the detection bounding box are shown in Appendix A
Figure A4.

The precision–recall curves of YoloV4 for pin and suspension disc insulators from different test datasets are shown in Figure 16. The average maximum precision is calculated with 11 recall values. Overall, the performance of the model is highly reliable when used in better light conditions.

#### 3.3.3. Performance Comparison with other Object Detection Models

To present the performance of the insulator detection model in comparison to other famous models, we trained YoloV5 [40] and its lighter version YoloV5s on the same normal insulator dataset that we used to train the YoloV4 architecture. The results presented in Table 4 show that the YoloV4 architecture outperforms YoloV5 in terms of AP for both normal pin and suspension disc insulators. Additionally, the YoloV5s model outperforms the YoloV4 Tiny model trained on our training dataset and evaluated on the same test datasets. YoloV4 remains as the best choice to further fine-tune on the specific faulty insulator dataset for better detection results.

Detection results of different object detection models along with bounding boxes are presented in Appendix A
Figure A4. Such an evaluation was performed to choose between the object detection models for further fine-tuning on the specific faulty insulator dataset.

#### 3.3.4. Faulty Insulator Detection using Proposed Methodology with Different Flight Path Strategies

To better inspect the defective porcelain insulators mounted on electric poles of overhead power distribution systems, we fine-tuned the insulator detection model in two stages. As stated earlier, the proposed fine-tuning strategy results are obtained with test datasets that are obtained using different flight path strategies for the UAV. The Prototype-C test dataset presents the frames taken from the video captured by the drone using a circular flight path. As stated in Table 5, an AP of 83.90% for faulty pin insulators and an AP of 70.56% for faulty disc insulators were achieved. With the proposed flight path, the dataset Prototype-S is captured and evaluated with the faulty insulator detection model. An AP of 56.51% on faulty pin insulators and an AP of 90.91% on suspension disc insulators are achieved. Overall F1-scores of 0.77 and 0.81 are obtained with Prototype-C and Prototype-S datasets, respectively.

The proposed straight flight path achieves a higher F1-score with the proposed fine-tuning strategy. Another major advantage of the straight flight path is less need for complex algorithms and less need for computation and components for obstacle avoidance. The PR curve is presented in Figure 17. Detection results, along with bounding boxes drawn from both of the datasets, are presented in Appendix A
Figure A5. After detecting the faulty insulator in a frame, the area is cropped from the image and the results are saved. Such a procedure will reduce the inspection time and allow the crew personnel to not need to observe the whole frame and they can check only the saved one for confirmation.

The proposed method presents a higher F1-score on insulator defect detection of overhead power line distribution systems. In [41], by using multi-task learning, an F1-score of 0.75 is achieved on insulator defect detection in overhead power lines, while our proposed methodology achieved an F1-score of 0.77 on the Prototype-C test dataset and an F1-score of 0.81 on the Prototype-S test dataset. Both of these datasets have both defective pin and suspension disc insulators.

Table 6 presents the F1-score of our proposed methodology for both faulty pin and disc porcelain insulators. The detection accuracy of our proposed methodology could possibly decrease when exposed to more complex case scenarios. For example, as shown in Figure A3a, such complex cases needed a more efficient fault detection algorithm. Further investigation is needed to design efficient flight path strategies when inspecting insulators on electric poles presenting high shuttering effects from clipping wires and conductors. Additionally, this study is limited to broken rain sheds, which are responsible for pollution deposition and surface corrosion. Further improvements can possibly be made for the detection of other types of faults present in porcelain insulators, i.e., cracks and damage, by enriching the training dataset with such faults.

## 4. Conclusions

In this paper, we present an approach to efficiently detect and recognize faults in porcelain insulators mounted on electric poles in primary distribution systems. Our approach deploys several image processing techniques to improve the quality of the training dataset, overcome the data insufficiency problem, and improve the generalization ability of the insulator detection model. Our results show that YoloV4 outperforms other state-of-the-art object detection models on our dataset. In addition, a less complex and more time- and energy-efficient flight path strategy for the UAV with our insulator fault detection model provides better detection and classification accuracy compared to those in previous literature. This study highlighted that the primary distribution system should be researched further. As future work, we will investigate the implementation of insulator fault detection, along with object avoidance algorithms and the detection of faults in insulators made of different materials. Additionally, we will extend the proposed approach to other power system components.

## Figures and Tables

**Figure 1 sensors-21-00974-f001:**
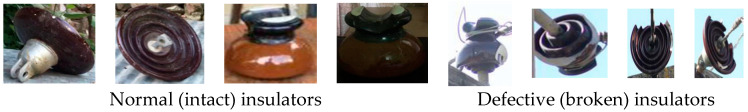
Depiction of insulators in overhead power distribution lines.

**Figure 2 sensors-21-00974-f002:**
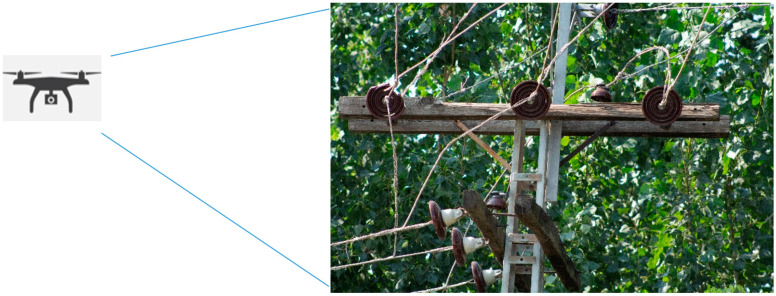
Illustration of overhead power distribution line from drone camera.

**Figure 3 sensors-21-00974-f003:**
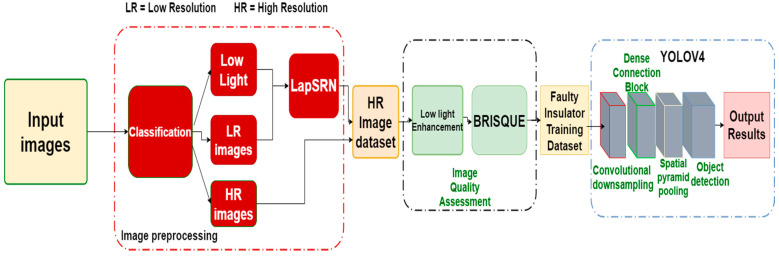
Workflow for the proposed methodology.

**Figure 4 sensors-21-00974-f004:**
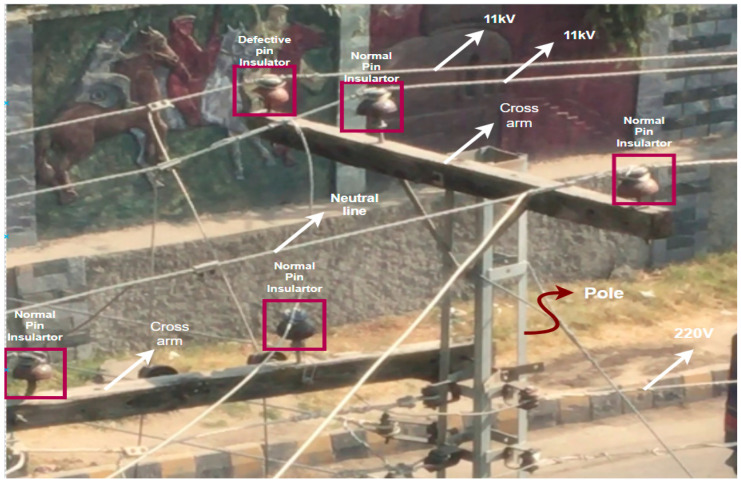
Power system components mounted on a supporting electric pole.

**Figure 5 sensors-21-00974-f005:**
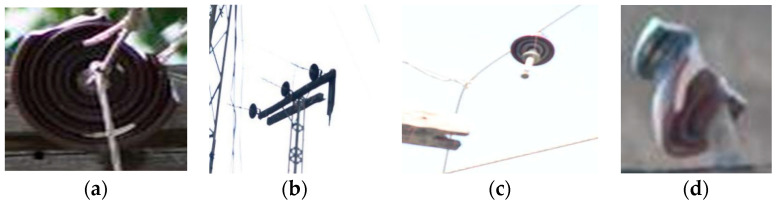
Different faults found in power distribution systems: (**a**) broken suspension disc insulator, (**b**) broken cross arm, (**c**) unclipped pin insulator from the cross arm, (**d**) broken pin insulator.

**Figure 6 sensors-21-00974-f006:**
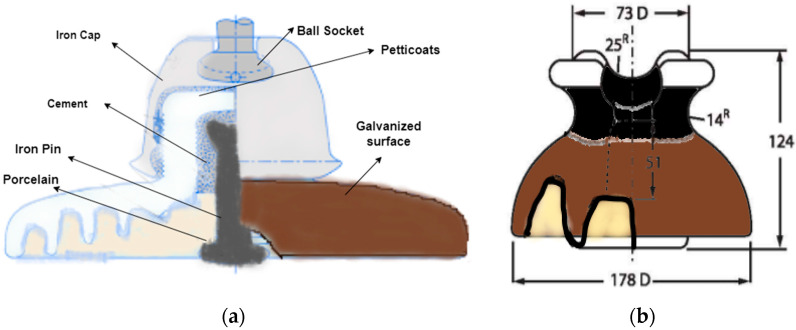
Internal structure of pin and suspension disc porcelain insulators: (**a**) internal structure of a suspension disc insulator, (**b**) the length parameters of the pin insulators.

**Figure 7 sensors-21-00974-f007:**
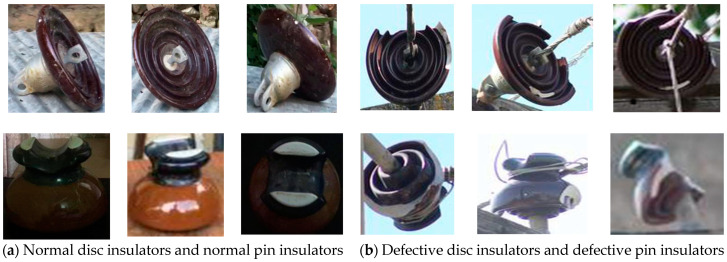
Normal (intact) and defective (broken) porcelain insulators with different view angles.

**Figure 8 sensors-21-00974-f008:**
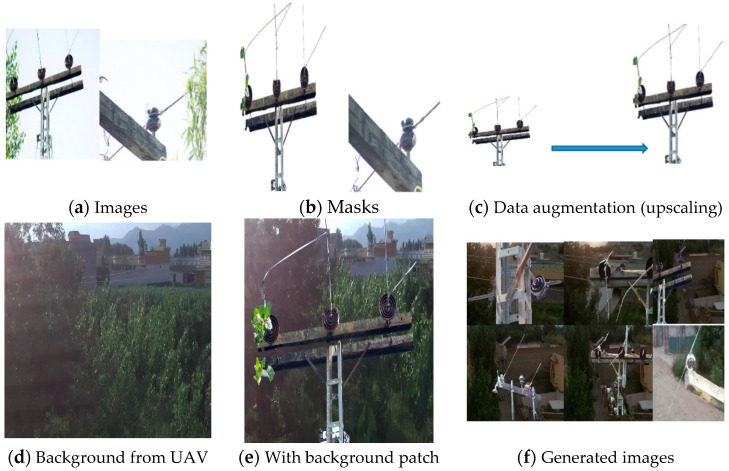
Steps of dataset formation from (**a**–**e**).

**Figure 9 sensors-21-00974-f009:**
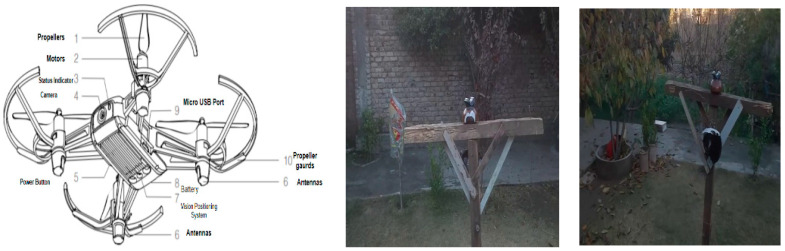
Components of the UAV and the prototype of the electric pole with broken insulators.

**Figure 10 sensors-21-00974-f010:**
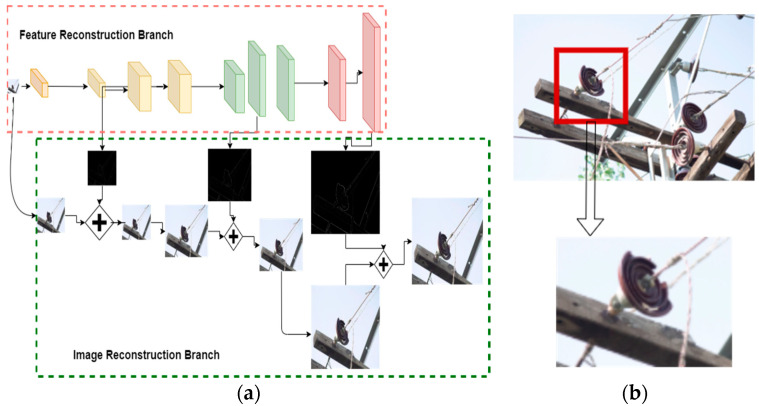
(**a**) Workflow of LapSRN for upsampling a low-resolution image, (**b**) LR patch extraction.

**Figure 11 sensors-21-00974-f011:**
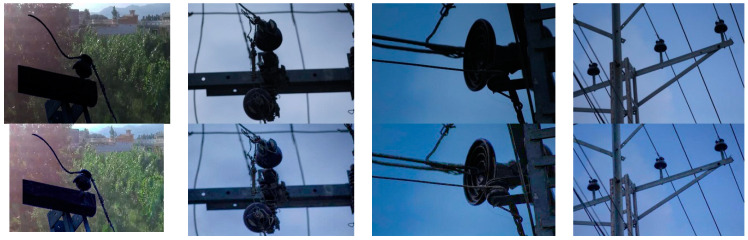
Low-light enhancement using LIME on different training images.

**Figure 12 sensors-21-00974-f012:**
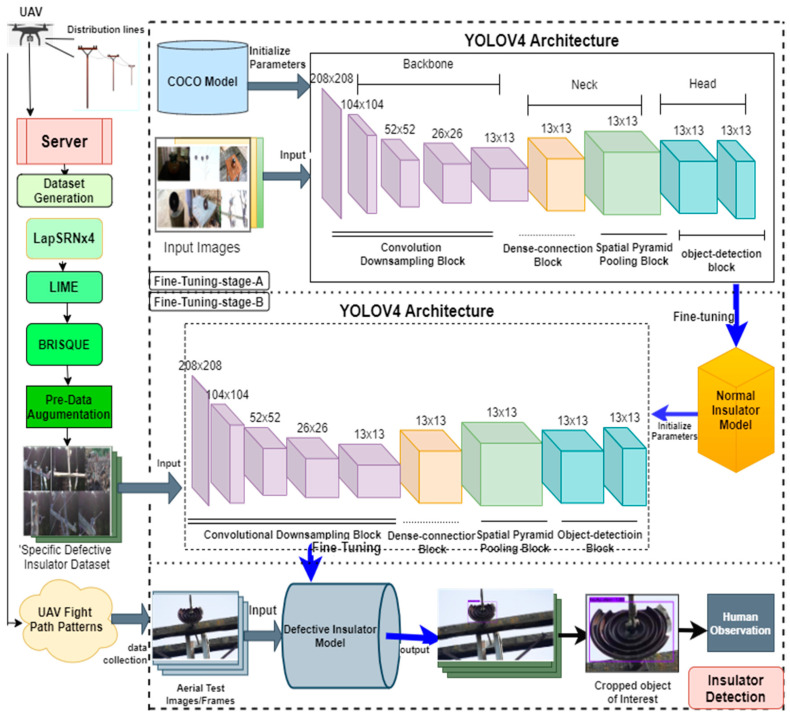
Fine-tuning strategy for insulator detection model.

**Figure 13 sensors-21-00974-f013:**
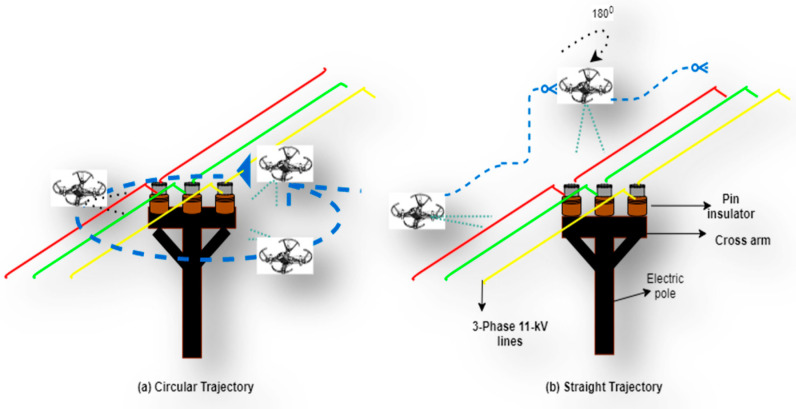
Flightpath strategies for the UAV for inspection of the insulators mounted on an electric pole. (**a**) Presenting a circular flight strategy (**b**) Presenting a straight flight path strategy.

**Figure 14 sensors-21-00974-f014:**
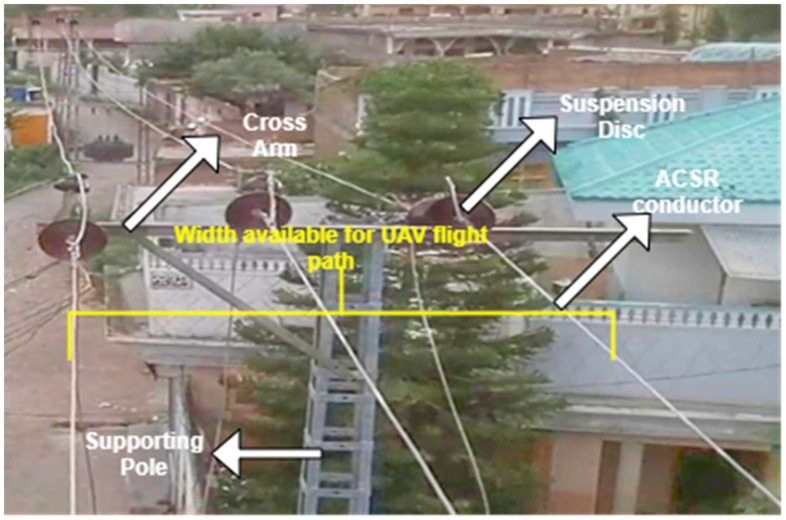
Aerial image of 3-phase 11kV lines showing the spacing between the conductor lines and obstacle-free path for the drone flight.

**Figure 15 sensors-21-00974-f015:**
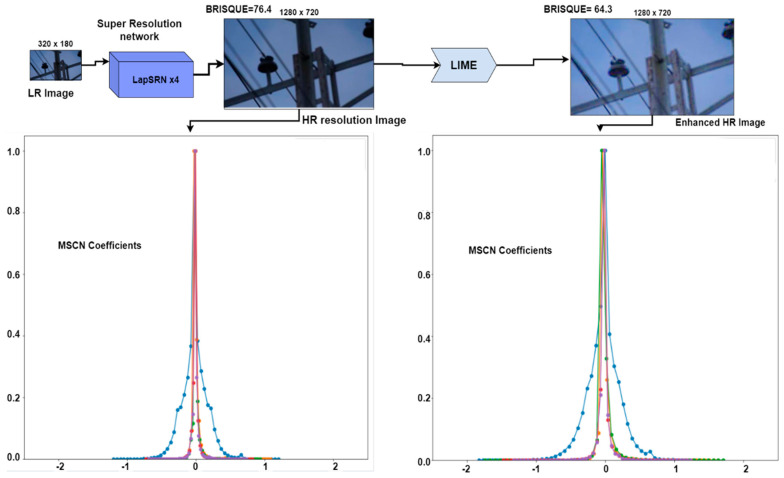
Improvement steps of specific insulator training dataset by LapSRN and LIME.

**Figure 16 sensors-21-00974-f016:**
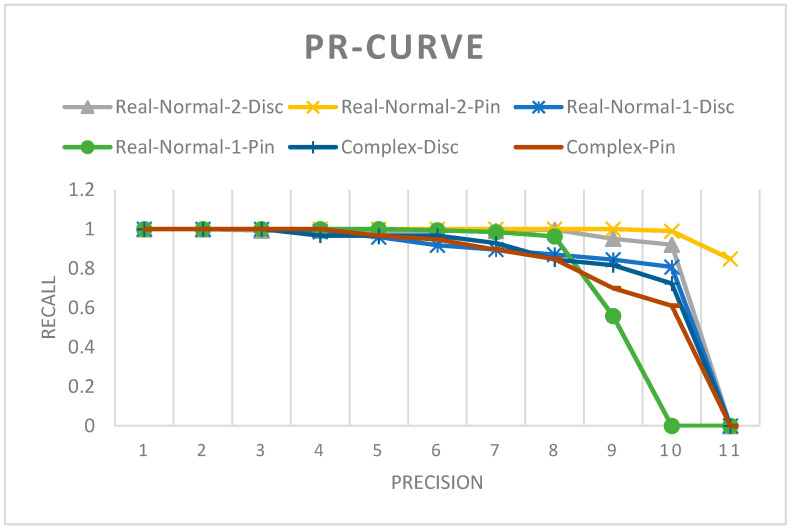
Precision recall curve of YoloV4 for normal insulator detection model for different test datasets.

**Figure 17 sensors-21-00974-f017:**
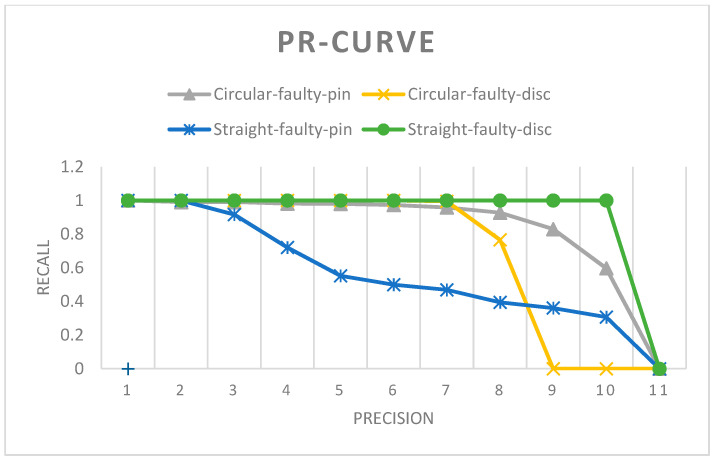
Precision vs. recall curve for the faulty insulator detection model with the proposed methodology.

**Table 1 sensors-21-00974-t001:** Technical specification of the UAV.

Technical Parameters	Value
Camera lens	FOV: 82.6° 25 mm
Takeoff weight	300 g
Video resolution	FHD: 1280 × 720 30 p
Max hovering time	15 min
Max flight speed	31 mph
Endurance	13 min
Positioning system	Vision

**Table 2 sensors-21-00974-t002:** Training and testing datasets for insulator detection model.

Training				Testing				
	Single-stage	Stage A	Stage B	R-1	R-2	C	Prototype-C	Prototype-P
Samples	5939	4827	4099	330	300	15	768	632

**Table 3 sensors-21-00974-t003:** Test results of YoloV4 and YoloV4 Tiny for different datasets.

Test Data Type	YoloV4 %AP50			YoloV4 Tiny %AP50		
	Pin	Disc	All Classes	Pin	Disc	All Classes
**R- 1**	78.2	87.7	***82.9***	44.4	72.6	58.5
**R- 2**	99.5	95.2	***97.4***	68.2	90.1	79.2
**Complex**	81.5	83.8	***82.7***	59.9	65.3	62.6

**Table 4 sensors-21-00974-t004:** Performance comparison of YoloV4 with other object detection models.

	YoloV4 %AP50			YoloV4 Tiny %AP50			YoloV5x %AP50			YoloV5s %AP50		
	Pin	Disc	All Classes	Pin	Disc	All Classes	Pin	Disc	All Classes	Pin	Disc	All Classes
R-1	78.2	87.7	***82.9***	44.4	72.6	58.5	58.4	78.6	68.5	54.8	78.7	66.8
R-2	99.5	95.2	***97.4***	68.2	90.1	79.2	99.4	88.3	93.9	96.4	87.3	91.9
Complex	81.5	83.8	***82.7***	59.9	65.3	62.6	77.3	70.7	74.0	87.3	55.2	71.2

**Table 5 sensors-21-00974-t005:** Results of proposed YoloV4 fine-tuning on faulty disc insulators using proposed flight path strategies.

Test Data Type	YoloV4 Proposed Methodology %AP50		
	Pin	Disc	F1-Score
Prototype-C	83.90%	70.56%	0.77
Prototype-S	56.51%	90.91%	0.81

**Table 6 sensors-21-00974-t006:** Results of the insulator detection model on prototype dataset with proposed flight path strategy.

Method		Insulator Defect Detection In OPDL
		F1-Score
Literature [41]		0.75
Proposed method	Prototype-C	***0.77***
	Prototype-S	***0.81***

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
