# Peer review of "Autonomous Vision-Based Primary Distribution Systems Porcelain Insulators Inspection Using UAVs"

_sensors, 2021, doi:10.3390/s21030974_

Round 1
Reviewer 1 Report
I have some experience in the field of Power Engineering and I am aware of the importance of fault detection in power line insulators, regardless of their type. I think that topics related to the automation of human labor, especially in hazardous environments is of significant scientific and social interest. In those lines, I consider the topic of the paper to be informative, modern and related to the current state of art with high quality of the scientific presentation.
Comments on the research methods
For the authors, I have the following comments on the research methods, which should not be taken as a necessary revision, but rather in my opinion, something that can be considered as an improvement or future work.
(1) In my opinion it would be beneficial if it is further elaborated on the fault detection, such as:
(a) is it possible for the algorithm to distinguish between different types of breakdowns;
(b) how the system reacts resolution wise to different breakdowns - such as distinguishing between complete breakage of a piece from the insulator against a small crack on its surface. In the training imagery provided in the paper
(c) what is the effect when different type of electric equipment is encountered – for example insulator with different material.
(2) It would be interesting to see how the particular techniques presented in the paper relate to other types of insulators – for example glass, which are prominent in my country of residence.
Spelling, grammar and formatting
As far as I can provide evaluation, as a nonnative English speaker, I find the text of the paper in terms of spelling, grammar and formatting to be of high quality, with only minor mishaps such as:
(1) The sentence on line 193 and 194 has the word need in to places that are five words apart.
(2) Line 230 – the caption of the figure is the next page.
(3) Line 267 – the brand name cannon should be with capital first letter.
(4) Line 307 – the reference is in Italic.
(5) Line 323 – the equation is missing its closing bracket.
References
I find the references sufficient adequate and informative.
Presentation
I have some minor issues with the graphical presentation:
(1) In some of the figures the borders are intersecting with the text (for example figure 12). It is not problem when reading the paper digitally but if its printed it might be difficult for some readers to see the text.
(2) The grid on Figure 16 and Figure 17 makes reading the figure difficult. Please consider increasing the opacity of the grid or removing it altogether.
Author Response
Dear Reviewer 1,
Greetings!
Thank you for reviewing our manuscript and presenting valuable suggestions and comments. Please see the attachment for point by point response to each comment.

Reviewer 2 Report
In this paper, a novel dataset is gathered along with different dataset generation techniques to overcome the data insufficiency problem. The quality of training images has been enhanced by the LapSRN algorithm and the less visibility problem is resolved by LIME. A different fine-tuning strategy is adopted by utilizing YoloV4 as a faulty insulator detection model. This technique allows the model to specifically learn the defective insulator features for more robust recognition and detection of faults in the aerial inspection. Comparison of different object detection model is also presented.
The authors could consider the following aspects in order to improve their work:
- The Introduction chapter is quite good and presents the state-of-the-art in the treated subject;
- The authors should provide more details related to the used mathematical approach/mathematical model which sustains the proposed method in this research;
- In the Conclusion chapter, the authors should present the obtained results based on the output data from experimental results; anyway, it will be interesting to make a comparison with other previous results on the same treated subject.
Author Response
Dear Reviewer 2,
Greetings!
Thank you for reviewing our manuscript and presenting valuable suggestions and comments. Please see the attachment for point by point response to each comment.
Best regards

Reviewer 3 Report
In this paper, a method for fault diagnosis of in-service porcelain insulators is presented based on real aerial inspection. The topic is interesting, and the organization of the paper is acceptable. However, there are some unclear items in the paper as well as some grammatical problems. Accordingly, the following issues should be addressed before the final decision.
- A comprehensive language revision is needed. For example, at the end of line 14, the sentence starts without capitalizing the first word. The same problem appears in line 165. Also, putting a point at the end of the title of a paper is not usual.
- The Abstract is too long and is another type of Introduction. It should briefly show the main objective of the paper and its findings.
- The authors should clearly mention their innovation compared to the existing methods in the literature. The items from line 166 to line 185 could not be considered as contributions.
- The structure of the paper should be included in the last paragraph of the Introduction section.
- The quality of some figures is too low (Figures 3 and 15 are the examples). Please replace them with higher quality ones.
- It seems that the authors just presented a technical report instead of a scientific article. The proposed model included in subsection 2.4 (Image reconstruction using LapSRN) contains limited mathematical formulations. What is the added value in this paper? The authors should provide an appropriate model, including more details about their contributions.
- The reviewer recommends making a comparison between periodic and non-periodic inspection methods. The comparison should give a direction for choosing the best method regarding the conditions and available devices.
Author Response
Dear Reviewer 3,
Greetings!
Thank you for reviewing our manuscript and presenting valuable suggestions and comments. Please see the attachment for point by point response to each comment.
Best regards.

Round 2
Reviewer 3 Report
The authors tried to address my comments. However, my third and fifth comments were not correctly covered.
Comment 3: The contributions should briefly show the novelties applied to the work. So, the first five contributions are too wordy. Also, the last item, "The proposed method has been evaluated with a test dataset comprised of different backgrounds, light conditions, and complex scenarios", could not be considered as a contribution. It is an aim, among others. Any paper is evaluated with different case studies or scenarios.
Comment 5: None of the numbers in the vertical either horizontal axes can be seen in Fig. 15 (MSCN Coefficients).
Author Response
Dear Reviewer 3,
Greetings!
Please see the attachment for point by point response to your sincere comments and suggestions.
Sincerely

Round 3
Reviewer 3 Report
I am happy with the answers. However, the modified version of the contributions should be replaced with the previous ones. Indeed, comment 3 should be addressed in the paper.
Author Response
Dear Reviewer 3,
Please see the attachment.
